# The Palladium(II) Complex of A*β*_4−16_ as Suitable Model for Structural Studies of Biorelevant Copper(II) Complexes of N-Truncated Beta-Amyloids

**DOI:** 10.3390/ijms21239200

**Published:** 2020-12-02

**Authors:** Mariusz Mital, Kosma Szutkowski, Karolina Bossak-Ahmad, Piotr Skrobecki, Simon C. Drew, Jarosław Poznański, Igor Zhukov, Tomasz Frączyk, Wojciech Bal

**Affiliations:** 1Institute of Biochemistry and Biophysics, Polish Academy of Sciences, 02-106 Warszawa, Poland; 1988.mariusz.m@gmail.com (M.M.); karolina.bossak@gmail.com (K.B.-A.); skrobec@gmail.com (P.S.); scdrew1@gmail.com (S.C.D.); jarek@ibb.waw.pl (J.P.); 2NanoBioMedical Centre, Adam Mickiewicz University, 61-614 Poznań, Poland; kosma.szutkowski@outlook.com

**Keywords:** Alzheimer’s disease, Aβ peptide, NMR spectroscopy, ^13^C relaxation, Palladium(II), ATCUN motif

## Abstract

The A$\beta_{4-42}$ peptide is a major beta-amyloid species in the human brain, forming toxic aggregates related to Alzheimer’s Disease. It also strongly chelates Cu(II) at the N-terminal Phe-Arg-His ATCUN motif, as demonstrated in A$\beta_{4-16}$ and A$\beta_{4-9}$ model peptides. The resulting complex resists ROS generation and exchange processes and may help protect synapses from copper-related oxidative damage. Structural characterization of Cu(II)A$\beta_{4-x}$ complexes by NMR would help elucidate their biological function, but is precluded by Cu(II) paramagneticism. Instead we used an isostructural diamagnetic Pd(II)-A$\beta_{4-16}$ complex as a model. To avoid a kinetic trapping of Pd(II) in an inappropriate transient structure, we designed an appropriate pH-dependent synthetic procedure for ATCUN Pd(II)A$\beta_{4-16}$, controlled by CD, fluorescence and ESI-MS. Its assignments and structure at pH 6.5 were obtained by TOCSY, NOESY, ROESY, ${}^{1}$H-${}^{13}$C HSQC and ${}^{1}$H-${}^{15}$N HSQC NMR experiments, for natural abundance ${}^{13}$C and ${}^{15}$N isotopes, aided by corresponding experiments for Pd(II)-Phe-Arg-His. The square-planar Pd(II)-ATCUN coordination was confirmed, with the rest of the peptide mostly unstructured. The diffusion rates of A$\beta_{4-16}$, Pd(II)-A$\beta_{4-16}$ and their mixture determined using PGSE-NMR experiment suggested that the Pd(II) complex forms a supramolecular assembly with the apopeptide. These results confirm that Pd(II) substitution enables NMR studies of structural aspects of Cu(II)-Aβ complexes.

## 1. Introduction

The A$\beta_{4-42}$ peptide is a major beta-amyloid species in the human brain. It was first co-discovered in 1985 as a major component of amyloid plaques of Alzheimer’s Disease (AD) brains, and was found to be more abundant than the commonly studied A$\beta_{1-42}$ and A$\beta_{1-40}$ peptides [1,2]. However, this discovery was soon disregarded, and the research was focused on the latter two peptides as forming neurotoxic aggregates. A$\beta_{4-42}$ was ignored for more than two decades, until new analytical methods based on immunochemistry coupled with mass spectrometry revealed A$\beta_{4-42}$ as the major Aβ species in various structures of healthy and AD brains [3,4]. A$\beta_{4-42}$ has been recently recognized as one of the fastest aggregating Aβ peptides and, consequently, proposed to be a key source of toxic amyloid deposits [5,6,7,8,9,10].

Aβ peptides possess a number of metal-binding amino acid residues, but they are concentrated in the N-terminal part of their sequence, including Asp1, Asp7, Glu3, Glu11, His6, His13 and His14. On the basis of chemical and biological studies, Cu(II) ions in particular have been implicated in the neurotoxicity of Aβ peptides. The logK for Cu(II) binding to A$\beta_{1-16}$ and A$\beta_{1-40}$ peptides at a physiological pH of 7.4 was determined as 10.0 and 10.1, respectively [11].

A large volume of work was devoted to the effects of Cu(II) on the Aβ aggregation and production of deleterious reactive oxygen species (ROS) via the Cu(II)/Cu(I) catalytic redox couple [12,13,14]. The bulk of this work was, consequently, devoted to A$\beta_{1-42}$ and A$\beta_{1-40}$ peptides, and their C-terminally truncated models spanning the Cu(II) coordination site, A$\beta_{1-28}$ and in particular A$\beta_{1-16}$ peptides. The relevance of ROS production by the Cu(II)/Cu(I) redox process enabled by A$\beta_{1-x}$ peptides was questioned, however, by the demonstration of a reverse relationship between the extent of A$\beta_{1-x}$ aggregation and toxicity and their ability to generate ROS via Cu(II) complexes [15].

On the other hand, Cu(II) complexes of A$\beta_{4-x}$ peptides are practically redox silent, as demonstrated in a ROS generation assay for A$\beta_{4-42}$ and its non-aggregating model A$\beta_{4-16}$ [16]. Moreover, the ascorbate activation ability of Cu(II)A$\beta_{4-16}$ is marginal compared to that of Cu(II)-A$\beta_{1-16}$ [17]. Furthermore, Cu(II)A$\beta_{4-x}$ form high-affinity Cu(II) complexes (log K = 13.5 and 14.2 at pH 7.4, for x = 16 and 9, respectively [16,18]). A$\beta_{4-16}$ was also able to withdraw Cu(II) ions from A$\beta_{1-16}$ immediately and quantitatively [16], but, in contrast with A$\beta_{1-x}$ peptides, strongly resisted copper transfer to metallothionein-3, except for under highly reducing conditions [19,20]. These properties prompted the concept that A$\beta_{4-42}$ may serve a physiological purpose in the maintenance of synaptic transmission as a Cu(II) scavenger, as reviewed recently [21]. Furthermore, its hydrolytic product, A$\beta_{4-9}$ is so inert to reduction by GSH that it may help shuttle Cu(II) ions across the blood–brain barrier [22].

These differences between the A$\beta_{1-x}$ and A$\beta_{4-x}$ peptides result from the different coordination modes they provide. The A$\beta_{1-x}$ peptides bind the Cu(II) ion in a heterogeneous fashion, using the N-terminal amine of Asp1, and various combinations of His6, His13 and His14 imidazole nitrogens, and donor atoms of the Asp1-Ala2 peptide bond [23]. In contrast, the A$\beta_{4-x}$ peptides take advantage of the truncation of the first three residues, which yields the so-called ATCUN/NTS motif, generated by Xaa-Yaa-His sequences, where the Cu(II) ion is coordinated to the N-terminal amine, the His imidazole and two intervening peptide nitrogens [24,25,26]. Although the Jahn-Teller effect in the $d^{9}$ electronic structure of the Cu(II) ion stipulates a tetragonal symmetry with axial ligand(s), these are usually weakly bonded when strong nitrogen ligands occupy equatorial positions. As a result, such axial sites are often unoccupied or occupied by solvent (water) molecules, as seen in relevant X-ray structures [27,28]. This situation has also been reproduced in the square planar coordination geometry of the Cu(II) complex of the Phe4-Arg5-His6 ATCUN/NTS motif obtained from DFT calculations [16].

Taking into account the abundance of the A$\beta_{4-42}$ peptide in the brain and its possible physiological role as a Cu(II) binding molecule, we considered that it would be very interesting to obtain a three-dimensional structure of the A$\beta_{4-16}$ model peptide in the Cu(II)-complexed form. Unfortunately, Cu(II) is a paramagnetic metal ion, and despite significant progress in NMR techniques, it is not possible to obtain detailed structural data for such a small complex [29,30].

The ATCUN/NTS motif yields practically isostructural complexes for Cu(II) and all other metal ions capable of displacing peptide nitrogens, including Ni(II) [31,32], Pd(II) and Au(III) [33]. These three metal ions have $d^{8}$ electronic configuration and their ATCUN/NTS complexes are square-planar and diamagnetic. Therefore, one of these metal ions can be used to substitute for Cu(II) in the A$\beta_{4-16}$ complex in order to perform an NMR structural study. Pd(II) is our best choice, for its ability to form very stable peptide complexes by displacing hydrogen atoms from the side chains as well as main-chain nitrogens even under acidic conditions [33,34]. As a consequence of the very high complex stability, they are also substantially more inert than the Cu(II) complexes, assuring the slow exchange condition in NMR experiments [34].

This paper describes structural and dynamic properties of the Pd(II) complex of the A$\beta_{4-16}$ peptide, investigated by NMR and additional techniques, including mass spectrometry, circular dichroism, and spectrofluorimetry. We demonstrate the usefulness of Pd(II) substitution for studies of the structure and reactivity of Cu(II) complexes of A$\beta_{4-x}$ peptides.

## 2. Results

### 2.1. Validation of the Pd(II) Complex as a Model for the Cu(II) Complex

Pd(II) is one of the most strongly hydrolysable cations, forming mononuclear Pd(OH)${}_{2}$ species at pH ∼ 1 and polynuclear forms at pH > 2 [35]. In order to avoid the kinetic entrapment of Pd(II) in such hydroxides en route to the desired Cu(II)-A$\beta_{4-16}$ mimic, we used K${}_{2}$PdCl${}_{4}$ as the Pd(II) source, which can resist hydrolysis and polymerization up to pH 5.5 at appropriate temperatures and concentrations [36,37,38]. In control experiments, we observed precipitation of a red-brownish condensate after 8–10 h for samples containing 1 mM $K_{2}{\mathrm{PdCl}}_{4}$ at pH 4.5. We, therefore, adopted a two-step preparation method of Pd(II)A$\beta_{4-16}$.

In the first step, the peptide was incubated at room temperature with substoichiometric (0.85 mol eq.) K${}_{2}$PdCl${}_{4}$ at pH 4.0, where the formation process of Pd(OH)${}_{2}$ and its polymerization to [Pd(OH)${}_{2}$-xClx]${}_{n}$ [39] was slower than the anchoring of Pd(II) to the peptide. The reaction was monitored using CD spectroscopy (Figure 1).

As shown in Figure 1, the initial anchoring of the Pd(II) ion to the peptide occurred promptly, as seen by an immediate appearance of chiral d−d bands in the spectra at the first time point of 5 min. This initial spectrum containing at least three bands is gradually superseded by a simpler one, containing two distinct d-d bands. The kinetics of formation of the latter has not been completed after 40 h. However, when NaOH was added to pH 6.5, the formation of a final reaction product was completed within about an hour and no hydroxide precipitation was observed. Therefore, we adopted a procedure in which the samples were incubated for 24 h at pH 4.0, followed by increasing the pH to 6.5, which was the standard value for most further experiments.

The samples prepared in this way were controlled by HPLC and major fractions were investigated by ESI-MS. Figure 2 presents an example of such data. Two major peaks contained uncoordinated A$\beta_{4-16}$ (at 27.2 min) and a monomeric Pd(A$\beta_{4-16}$) complex (at 37.7 min). Several minor peaks also contained the complex of the same stoichiometry. Both the peptide and the complex exhibited three charge species, 2+, 3+ and 4+ in electrospray ionisation mass spectrometry (ESI-MS) spectra, with roughly similar proportions of peak intensities. Figure 2 also presents the ion-mobility spectrometry–mass spectrometry (IMS-MS) spectra of 2+ and 3+ charge species of A$\beta_{4-16}$ and Pd(A${}_{4-16}$), showing very little difference in drift times between the peptide and its Pd(II) complex.

The next experiment compared the effect of Pd(II) and Cu(II) ions on Tyr10 fluorescence, a common tool in studying metal binding to Aβ peptide [11,16]. In order to maintain a coherence between the Pd(II) and Cu(II) complexes, the samples were prepared in a different way. The required aliquots of stock solutions of Cu(II) salt were added directly to individual 25 μM A$\beta_{4-16}$ samples dissolved in 20 mM MES buffer at pH 6.5, or 20 mM HEPES at pH 7.4. The Pd(II) samples were prepared using a two-step procedure: initially, appropriate Pd(II) solutions with A$\beta_{4-16}$ peptide were prepared in water, at pH 4, and incubated for 48 h. Next, the appropriate buffer was added to achieve a final solution with 20 mM MES, pH 6.5, or 20 mM HEPES, pH 7.4. The resulting titration curves, either for pH 6.5 (Figure 3) and 7.4 (Appendix A) indicate that both metal ions quench Tyr10 fluorescence in a similar stepwise manner. The straight lines in Figure 3 and Appendix A are linear fits to the titration curve segments corresponding to the binding of the metal ions at the 1st and 2nd binding sites, demonstrated for Cu(II) ions [16], and presumably equally valid for Pd(II).

The results of the above experiments, taken together, strongly supported the assumption that Pd(II) may be an isostructural substitute for Cu(II) in the ATCUN/NTS motif of A$\beta_{4-16}$ for structural studies, forming a mononuclear 1:1 species at pH 6.5 and 7.4. Therefore, we performed a series of NMR experiments aimed at elucidating the solution structure of this complex.

### 2.2. Structural Analysis of the A$\beta_{4-6}$ Peptide as a Simple Model of the ATCUN/NTS Site in A$\beta_{4-16}$

The short Phe-Arg-His-amide (FRH) peptide represents the N-terminal residues forming the Pd(II) binding ATCUN/NTS motif. For structural and dynamic aspects of Pd(II) binding, two FRH samples—without the metal (apo) and saturated with Pd(II)—were used to perform homo- and heteronuclear NMR experiments. The acquired experimental data enabled the assignment of all ${}^{1}$H and ${}^{13}$C resonances in both (apo and Pd(II) saturated) forms (Appendix A). The ${}^{1}$H-${}^{13}$C HSQC spectra recorded for the apo A$\beta_{4-6}$ and Pd(A$\beta_{4-6}$) are presented in Figure 4. The amide ${}^{1}$H${}^{N}$ signals of Arg5 and His6 were not detected due to fast exchange with water in the apo form, and coordination of the Pd(II) binding in the complex. In the ${}^{1}$H-${}^{15}$N HSQC spectrum, only the signals from the NH${}_{2}$ group at the Phe4 N-terminus were detected (Appendix A). Large downfield chemical shifts along the ${}^{13}$C axis were observed for ${}^{13}$Cα signals for Phe4 and Arg5, together with and ${}^{1}$H shift for ${}^{13}$Cα–${}^{1}$Hα in His6 (Figure 4A) confirmed that backbone amide groups facilitate the coordination sites for Pd(II) ion. The fourth site was determined from the analysis of the aromatic part of the ${}^{1}$H-${}^{13}$C HSQC spectrum (Figure 4B) and selected as ${}^{1}$Hϵ1 proton in His6. The geometric parameters of the FRH complex with Pd(II) were extracted from the high-quality 3D structure of GGH tripeptide with Pd(II) ion [33]. Finally, the high-resolution 3D structure of FRH–Pd(II) complex (Figure 5) was solved by refined initial structure in water box with the YASARA software [40].

To obtain information on molecular dynamic processes upon Pd(II) binding, the ${}^{13}$C relaxation rates ($R_{1}$ and $R_{2}$) were measured for the ${}^{13}$C resonances in Phe4 and His6 aromatic side-chains (Appendix A). The ${}^{13}$C${}^{\delta }$ and ${}^{13}$C${}^{\epsilon }$ resonances in the Phe4 side-chain did not reveal differences between the apo and Pd(II) cases, suggesting that they were not affected by Pd(II) binding. In contrast, the ${}^{13}$C resonances in the His6 side-chain demonstrated substantial differences between both states, confirming the formation of the coordination bond in the His6 imidazole ring. The $R_{2}$ relaxation rate for ${}^{13}$C${}^{\delta 2}$ was significantly decreased by Pd(II). At the same time, the Pd(II) binding stimulated the increase of $R_{1}$ relaxation rate for both His6 resonances–${}^{13}$C${}^{\epsilon 1}$ and ${}^{13}$C${}^{\delta 2}$ (Appendix A). Taking into account that the $R_{2}$ relaxation rate reflects the intensity of the dynamic processes in the low-frequency time frame (ms–μs), we can conclude that Pd(II) binding resulted in shifting the molecular dynamics processes from the ms–μs to the μs–ns regime (Appendix A).

### 2.3. Solution Structure of the Pd(II) Complex with the A$\beta_{4-16}$ Peptide

The A$\beta_{4-16}$ peptide together with the Pd(A$\beta_{4-16}$) complex were subjected to structural analysis in solution based on NMR data. The combination of homonuclear and heteronuclear NMR spectra yielded the assignments of more than 95% of ${}^{1}$H, ${}^{13}$C and ${}^{15}$N resonances in both forms (See Appendix A). In fact, in the apo A$\beta_{4-16}$ peptide, only amide protons in the three N-terminal residues were not observed, due to fast exchange with water protons. The resonances of His14 were not assigned due to degeneration of signals from His13 (Appendix A). For the Pd(A$\beta_{4-16}$) complex, the analysis of NMR data enabled us to assign practically all resonances, except for the amide protons of Phe4, Arg5 and His6, together with the H${}^{\delta 1}$ proton in His6, which were displaced upon Pd(II) binding (Appendix A).

Neither the A$\beta_{4-16}$ peptide nor the Pd(A$\beta_{4-16}$) complex yielded a substantial amount of nontrivial long- and medium-distance constraints in homonuclear NOESY or/and ROESY experiments. Therefore, the 3D structure of apo A$\beta_{4-16}$ in solution was evaluated with the Xplor-NIH (version 2.39) software mostly on the basis of backbone ϕ and ψ torsion angles, deduced from chemical shifts with the TALOS-N program [41]. The ensemble of 20 low-energy structures after additional refinement with the explicit solvent model demonstrated the existence of some structuring only in the His6–Tyr10 region (Appendix A).

The 3D structure of the complex A$\beta_{4-16}$ with an equimolar amount of Pd(II) reveals metal coordination according the ATCUN/NTS motif (Figure 6). Performed structural analysis suggest that Pd(II) bind to A$\beta_{4-16}$ in the same manner as to the Pd(A$\beta_{4-6}$). In comparison to the apo form, the equimolar complex of Pd(II) resulted in the small chemical shift perturbations (csp) detected for amide ${}^{1}$H${}^{N}$ protons for the residues in ${}^{7}$DSGYEVHHQK${}^{16}$ fragment of the A$\beta_{4-16}$ peptide. There are two sets signals observed for Tyr10 (Appendix A), which suggests this residue exists in two conformations.

### 2.4. Translational Mobility of A$\beta_{4-16}$ and Pd(A$\beta_{4-16}$) in Solution

The translational mobilities of the Pd(A$\beta_{4-16}$) complexes in solution were studied by diffusion measurements obtained at 11.7 T. The experimental data were analyzed on the basis of the Stejskal-Tanner equation [42]:$$I=I_{0}\exp\left(-D{\left(G\gamma \delta \right)}^{2}\left(\Delta -\delta /3\right)\right)$$
where $\gamma_{H}$ is the ${}^{1}$H gyromagnetic ratio, δ is the gradient duration, Δ is the diffusion time and *G* is the gradient strength. The coefficient of translational diffusion ($D_{tr}$) in solution was obtained for A$\beta_{4-16}$ peptide in apo form and for two concentrations of Pd(II) ions (Figure 7). The $D_{tr}$ value obtained for the apo-peptide is 1.64 ± 0.01 × 10${}^{-10}$ (m${}^{2}$/s), for the equimolar ratio in the Pd(A$\beta_{4-16}$) complex, the measured $D_{tr}$ was 1.65 ± 0.02 × 10${}^{-10}$ m${}^{2}$/s which corresponds to an effective hydrodynamic volume 7.42 (nm${}^{3}$). The Pd(A$\beta_{4-16}$) complex 1:1.4 characterized $D_{tr}$ equal to 1.34 ± 0.02 × 10${}^{-10}$ m${}^{2}$/s, corresponding to the effective hydrodynamic volume 15.98 (nm${}^{3}$), which is more than two times higher compared to apo-A$\beta_{4-16}$. The apparent hydrodynamic volume was calculated according to the Stokes–Einstein equation using a spherical approximation. Although the morphology of the diffusing species is not spherical, the proposal approximation is widely used to control the aggregation phenomena [43,44].

## 3. Discussion

The CD and fluorescence spectroscopic and MS data strongly suggested that the Pd(II) ion accommodated the same ATCUN/NTS coordination structure in A$\beta_{4-16}$ as Cu(II). The evidence was indirect, however. For CD (Figure 1), it was based on the characteristic alternate pattern of Pd(II) d−d bands, blueshifted, compared to the Cu(II) case, but retaining their symmetry [16]. For Tyr10 fluorescence, the same quenching pattern was seen for both metal ions (Figure 3), with the quenching just slightly more effective for the first Pd(II) equivalent. The correct stoichiometry indicating the replacement of four hydrogens by Pd(II) coordination was provided by ESI-MS (Figure 2A). In addition, the overlapping of the IMS drift peaks for the apo peptide and the complex is consistent with this view, because the lack of drift time difference indicates that the Pd(II) ion did not produce the long-range structuring in the molecule, and hence it had to be coordinated locally at one end of the molecule (Figure 2B).

These observations prompted the NMR study of the structure and dynamics of the Pd(A$\beta_{4-16}$) complex. However, while the observations presented above strongly suggested that the Pd(II) ion was bound selectively at the ATCUN/NTS Phe4-Arg5-His6 sequence, some binding at His13 and/or His14 residues could not be excluded *a priori*. Therefore, structural experiments were also performed using the Phe-Arg-His-amide tripeptide. This peptide has essentially one possible Pd(II) binding mode at weakly acidic pH, namely the ATCUN/NTS four-nitrogen complex, as evidenced by the X-ray study of its analogue Gly-Gly-His [33]. It could, therefore, serve as a positive control for the A$\beta_{4-16}$ experiments. The analysis of NMR spectra of the tripeptide complex, presented in Figure 4 and Appendix A confirmed the expected coordination mode. Very interestingly, the Pd(II) complexation did not affect the dynamics of the Phe4 side chain, but significantly elevated the longitudinal and decreased the transverse ${}^{13}$C relaxation rates for the His imidazole ring. This observation is in line with the formation of the six-membered chelate ring formed by simultaneous coordination of His amide and His N1 nitrogens.

The experiments performed for the equimolar Pd(A$\beta_{4-16}$) sample confirmed the Pd(II) coordination in the 4N mode to the Phe-Arg-His sequence, and the lack of longer distance structuring in the coordinated peptide, as shown in Figure 6. In particular, there was no interaction between Tyr10 and the Pd(II) coordination site, which was previously detected for a generally similar Ni(II) complex with the N-terminal peptide of HP2 protamine [45]. In the structure of the Pd(II) binding site in the tripeptide Phe-Arg-His (A$\beta_{4-6}$) presented in Figure 5, no interactions between the Phe, Arg and His side chains are present. These side chains are located away from each other. In particular, there is no axial interaction of Phe4 π charge with the axial electronic density of the Pd(II). Such interaction was seen before in the NMR structures of di- and tripeptides containing a Tyr residue [46,47], but apparently is not sufficiently effective for the less polarized Phe aromatic ring. The positively charged guanidinium group of Arg5 appears to be fixed in its position by electrostatic interaction with the partial negative charge located on Pd(II)-coordinated amide nitrogens [45]. The Phe4 and Arg5 positions are also similar to those calculated previously for the Cu(II) complex with the A$\beta_{4-7}$ fragment by DFT [16]. The same structure was retained in Pd(A$\beta_{4-16}$). The hydrogen bond between Phe4 and Asp7 postulated by previous DFT calculations was not detected.

The mobility of Pd(A$\beta_{4-16}$) was studied in comparison to the apo-peptide, and also in the presence of the apo-peptide excess. Interesting observations were made in these experiments. In agreement with the IMS data, the Pd(II) coordination did not affect the peptide’s hydrodynamic radius, thereby confirming the purely local character of the complexation on the peptide structure. A significant decrease in the complex mobility was observed, however, in the sample containing the complex and a 40% excess of the peptide. The apparent hydrodynamic radius calculated for this sample was about two-times larger than that for either the apo-peptide or the complex. This effect can be only explained by the supramolecular assembly between the complex and the apo-peptide. The absence of changes in chemical shifts of the Phe-Arg-His sequence upon apo-peptide addition indicates that no coordinative bridging by Pd(II) occurred. Two possible mechanisms for this effect can be considered. As the square-planar Pd(II) chelate structure is the only permanent and significant difference between the complexed peptide and the apo-peptide, one possible interaction involves its dipolar interaction with aromatic residues of the apo-peptide, similar to the interaction observed between the isoelectronic Ni(II) complex of the Arg-Thr-His N-terminus and the Tyr phenol ring in the HP2 pentadecapeptide [45]. In general, many types of stacking interactions that have been observed in various Cu(II) and Pd(II) biomimetic complexes might occur in the investigated system [48]. Another may stem from electrostatics, as proposed recently in studies of Cu(II) effects on fibrillization of A$\beta_{1-40}$ and A$\beta_{4-40}$ peptides [49]. The electrostatic charges of A$\beta_{4-16}$ and Pd(A$\beta_{4-16}$) at pH 6.5 can be calculated from the data presented before for the Cu(II) complex, reasonably assuming that the Cu(II)/Pd(II) replacement did not affect the acidities of the peptide’s residues. The average charges of these two species are +1.1 and +0.5, respectively. If, however, we consider the +1 charge at the C-terminal Lys residue in both molecules, then we see that there is a slight electrostatic incentive for the attractive interaction specifically between the apo-peptide and the complex spread over the rest of the molecule. Probably, both kinds of interactions occur, enabling the formation of the heterodimer or even higher order assemblies, even despite the lack of defined structure in free and Pd(II)-complexed A$\beta_{4-16}$. These results confirm the suitability of Pd(II) substitution to study structural aspects of Cu(II) complexes of Aβ peptides and suggest that the molecular pathway of their aggregation processes may lead via interactions of their N-termini.

## 4. Materials and Methods

### 4.1. Materials

The A$\beta_{4-16}$ (FRHDSGYEVHHQK-amide) and A$\beta_{4-6}$ (FRH-amide) peptides were synthesized according to Fmoc strategy as described previously [16]. K${}_{2}$PdCl${}_{4}$, NaOH, HCl and acetonitrile (HPLC grade) were purchased from Sigma-Aldrich. D${}_{2}$O was purchased from Armar Chemicals.

### 4.2. Sample Preparation

A portion of lyophilized A$\beta_{4-16}$ peptide was diluted in MQ water. The concentration of this stock solution was determined using $\epsilon_{275}$ = 1375 M${}^{-1}$cm${}^{-1}$ [16]. Pd(II) was added to the sample from a 100 mM K${}_{2}$PdCl${}_{4}$ stock of to obtain a peptide-to-metal ratio of 1:0.85. The pH of the sample was then increased by adding small amounts of concentrated NaOH up to pH 4, and then handled further, as required by specific experimental methods.

### 4.3. Mass Spectrometry

The samples prepared as above, containing 0.5 mM A$\beta_{4-16}$ and the peptide-to-Pd(II) ratio of 1:0.85 were incubated for a further 24 h at pH 4. Then, the pH was set to 6.5 using concentrated NaOH. 100 μL samples were injected into the HPLC system (Empower, Waters), equipped with an analytical C18 column (4.6 × 250 mm). The eluting solvent A was 0.1% (*v/v*) TFA in water, and solvent B was 0.1% (*v/v*) TFA in 90% (*v/v*) acetonitrile. The chromatograms were obtained at 220 and 280 nm. Individual peaks were collected and measured by ESI-MS on a ESI Q-ToF Premier mass spectrometer (Waters). The samples were injected at a 40 mL/min flow rate and MS spectra were recorded in positive ion mode during 5 min in the range m/z of 500–1800. Obtained mass spectrometry data were analyzed and processed using MassLynx (Version 4.1, Waters Inc., Milford, MA, USA). Ion mobility (IMS-MS) experiments were performed using a Synapt G2 HDMS instrument (Waters). Ions were generated using nanoelectrospray ionization at 1.7 kV from PicoTip emitters 2 μm i.d. (QT10-70-2-CE-20 New Objective). MS settings were adjusted to obtain an optimal ion transmission as follows: 30 V sampling cone, 5 V extractor cone, 40 ${}^{\circ }$C source temperature, 10 V trap collision energy, and 5 V transfer collision energy, wave height and wave velocity were set as 40 V and 800 m/s, respectively. Drift times were obtained by generating an extracted ion chromatogram (XIC) from the arrival time distribution function in MassLynx v4.1 using the monoisotopic mass and a mass window of ±0.075 Da.

### 4.4. Circular Dichroism

CD experiments were carried out on the J-815 CD spectrometer (JASCO) over the spectral range of 250–600 nm, using a 1 cm path length quartz cuvettes. Measurements were performed at 25 ${}^{\circ }$C for samples containing 0.3 mM A$\beta_{4-16}$ and the peptide-to-Pd(II) ratio of 1:0.85. The spectra were recorded in 40 min intervals, starting immediately after sample preparation, until the reaction neared equilibrium after 40 h. Next, the pH was set to 6.5 using NaOH and spectral changes were recorded at 30 min intervals for another six hours.

### 4.5. Spectrofluorimetry

Fluorescence spectra of A$\beta_{4-16}$ in the presence of Cu(II) and Pd(II) ions were recorded at 25 ${}^{\circ }$C using a FP-6500 spectrofluorometer (Jasco). The excitation wavelength was 280 nm; the emission spectra were in the range of 290–400 nm. Solutions of metal ions and the peptide were combined in varying metal/peptide ratios, with a constant A$\beta_{4-16}$ concentration of 25 μM, in 20 mM MES buffer, pH 6.5, or 20 mM HEPES buffer, pH 7.4. Cu(II) salt solution was added directly to the prepared peptide solution in the respective buffer. Pd(II) was added to water solutions of the peptide at pH 4, then incubated for 48 h. Next, to such solutions, appropriately concentrated buffer solutions were added to obtain concentrations analogous to Cu(II) samples. Each sample was prepared in triplicate.

### 4.6. NMR Spectroscopy

The Pd(II)-containing samples were prepared by incubation for 24 h at room temperature at pH ≈ 4.0. Next, the samples were diluted with D${}_{2}$O to obtain a 10% (*v/v*) solution of the latter, followed by adjustment of pH to 6.5 with a small amount of concentrated NaOH. The samples of the apo A$\beta_{4-6}$ and A$\beta_{4-16}$ peptide was prepared directly in 90%/10% (*v/v*) H${}_{2}$O/D${}_{2}$O, at pH 6.5. The final volumes of the samples were 300 μL, with final peptide concentrations between 1.0 and 2.5 mM.

The measurements were conducted on Agilent DDR2 800 (${}^{1}$H frequency 799.903 MHz), Agilent DDR2 600 (${}^{1}$H frequency 599.930 MHz), and Varian Inova 500 (${}^{1}$H frequency 500.606 MHz) NMR spectrometers operated at magnetic fields of 18.8 T, 14.1 T, and 11.7 T, respectively. All spectrometers were equipped with three channels, *z*-gradient unit and ${}^{1}$H/${}^{13}$C/${}^{15}$N probehead with inverse detection. The homonuclear 2D NMR experiments included TOCSY acquired with mixing times of 15, 80, and 90 ms, ROESY conducted with the mixing time of 300 ms, and NOESY with mixing times of 150 and 300 ms. Homonuclear experimental data sets were supplemented with heteronuclear 2D ${}^{1}$H-${}^{13}$C HSQC (tuned independently to aliphatic and aromatic regions) as well as 2D ${}^{1}$H-${}^{15}$N HSQC NMR experiments acquired using the natural abundance of ${}^{13}$C and ${}^{15}$N isotopes. All NMR data sets were referenced indirectly in respect to external sodium 2,2-dimethyl-2-silapentane-5-sulfonate (DSS) with the Ξ coefficients equal to 0.251449530 and 0.101329118 for ${}^{13}$C and ${}^{15}$N dimensions, respectively [50]. The acquired data sets were processed with the NMRPipe program [51] and analyzed with the Sparky software [52].

### 4.7. Assignment of the ${}^{1}$H, ${}^{13}$C and ${}^{15}$N Resonances and 3D Structure Evaluation of A$\beta_{4-16}$ Peptide and Pd(A$\beta_{4-16}$) Complex in Solution

The ${}^{1}$H, ${}^{13}$C and ${}^{15}$N resonance assignments were obtained using base standard procedure on base analysis 2D homonuclear (NOESY, TOCSY) and heteronuclear (${}^{1}$H-${}^{13}$C and ${}^{1}$H-${}^{15}$N HSQC) spectra. More than 90% of resonances were successfully assigned (Appendix A). Unfortunately, the collected 2D NOESY and ROESY data sets did not provide nontrivial medium- or long-range distance constraints. Nevertheless, the analysis of ${}^{1}$H, ${}^{13}$C and ${}^{15}$N chemical shifts performed with the TALOS-N software [41] yielded 16 and 8 backbone ϕ and ψ torsion angles predicted for A$\beta_{4-16}$ peptide in apo and Pd(II)-saturated forms, respectively. Additionally, the χ1 torsion angles of the side chains were predicted in six and four residues in A$\beta_{4-16}$ peptide and Pd(A$\beta_{4-16}$) complex, respectively (Appendix A). Both 3D structures were solved by the Xplor-NIH 2.37 program [53]. The slightly modified standard protocol included in Xplor-NIH distribution (protG.inp) was used for 3D structure evaluation. Briefly, the 200 randomly generated structures which were subjected to 12 ns cartesian molecular dynamic simulation at 2000 K followed by 3000 steps of simulating annealing included slow cooling to the temperature 100 K. The refined procedure in the explicit solvent were performed for the 20 lowest-energy structures with the YASARA software utilizing AMBER14 force field during the 3000 steps of simulating annealing procedure in water solvent [40]. In the case of Pd(A$\beta_{4-16}$) complex, the parameters for Pd(II) ion (bond length and angles) were taken from the crystal structure of Pd(II) complex with the GGH tripeptide [33]. The evaluated 3D structures were visualized and analyzed with the MOLMOL [54] and Chimera [55] software.

### 4.8. ${}^{13}$C Relaxation Measurements

The ${}^{13}$C $R_{1}$ and $R_{2}$ relaxation data were acquired on Varian Inova 500 NMR spectrometer for FRH peptide in both (apo and Pd(II) saturated) at natural abundance of ${}^{13}$C isotope. The experiments were conducted utilizing the pulse sequence included in BioPack (Agilent Inc., PaloAlto, CA, USA). The 32 scans were enabled to achieve a reasonable signal-to-noise ratio for collected points in $R_{1}$ and $R_{2}$ measurements. The $R_{1}$ relaxation data were obtained as 5 delays—10, 50, 110, 190, and 290 ms. The $R_{2}$ relaxation rates were recorded with 5 delays—10, 30, 50, 70, and 90 ms. The recycling delay was set to 3 s.

### 4.9. Diffusion Measurements

Diffusion experiments were carried out on a Varian Inova 500 NMR spectrometer utilizing a standard PGSE (Pulsed Gradient Spin Echo) pulse sequence [56] supplemented with Excitation sculpting Solvent Suppression block [57] were applied as 25 gradients with an effective gradient pulse duration (δ) as long as 3 ms. The diffusion measurements for A$\beta_{4-16}$ saturated with Pd(II) were performed using diffusion time (Δ) of 100 and 150 ms in the case of Pd(II) concentrations 1 and 1.4. The 128 accumulations were performed with a relaxation delay of 3 s in order to increase the signal-to-noise ratio. The obtained experimental data were Fourier transformed with a 2 Hz line broadening factor applied. For extraction of the translation diffusion coefficient ($D_{tr}$), the signals observed between 0–3 ppm were integrated and then exported to the Origin software together with gradient amplitudes. The $D_{tr}$ values were extracted by fitting using the Stejskal–Tanner equation [42] taking into account an additional correction for the Δ delay during BPP pulse in sequence (delay between gradient δ and π/2 pulse was equal to 0.5 ms).

## 5. Conclusions

Our study demonstrated that by Pd(II) substitution working models of Cu(II) complexes of ATCUN/NTS peptides, including those of the A$\beta_{4-x}$ family, suitable for direct structural and functional studies can be obtained. Already in this pioneering study, the diffusion coefficient determination, essentially impossible for the paramagnetic Cu(II) complex empowered us to confirm a supramolecular interaction contributing to the control of aggregation and fibrillization of the Aβ peptides. This interaction is a prerequisite for a better understanding of the molecular events leading to Alzheimer’s disease and thereby finding key markers of the disease.

## Figures and Tables

**Figure 1 ijms-21-09200-f001:**
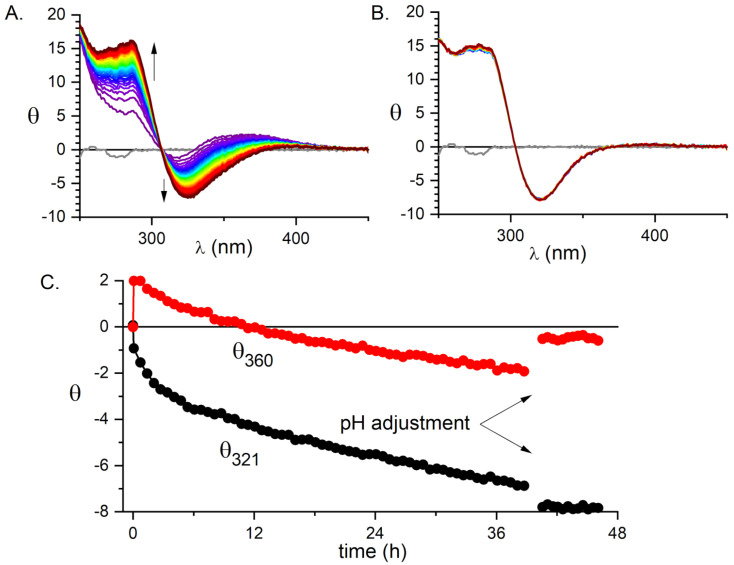
The formation of the 1:1 Pd(II) complex of A$\beta_{4-16}$ monitored by CD. (**A**) The 40 h evolution of CD spectra of the sample containing 0.3 mM A$\beta_{4-16}$ and 0.255 mM $K_{2}{\mathrm{PdCl}}_{4}$ at pH 4.0 and room temperature. (**B**) The six-hour evolution of CD spectra of the same sample after the change of pH to 6.5. (**C**) The time dependence of ellipticity at 360 and 321 nm.

**Figure 2 ijms-21-09200-f002:**
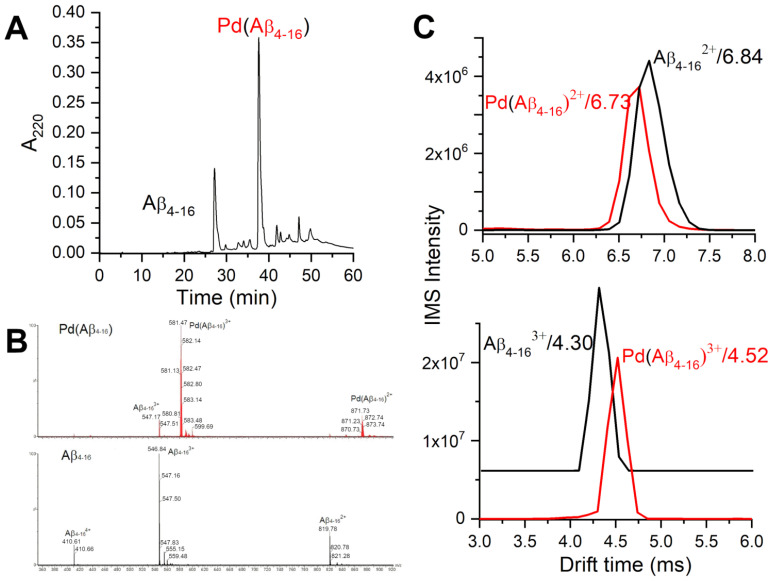
HPLC/MS characterization of products of the two-step procedure of obtaining the Pd(A$\beta_{4-16}$) samples. (**A**) HPLC chromatogram of the sample containing initially 0.5 mM A$\beta_{4-16}$ and 0.425 mM K${}_{2}$PdCl${}_{4}$. (**B**) ESI-MS spectra of the respective HPLC peaks. (**C**) IMS characterization of A$\beta_{4-16}$ and Pd(A$\beta_{4-16}$) species at pH 6.5.

**Figure 3 ijms-21-09200-f003:**
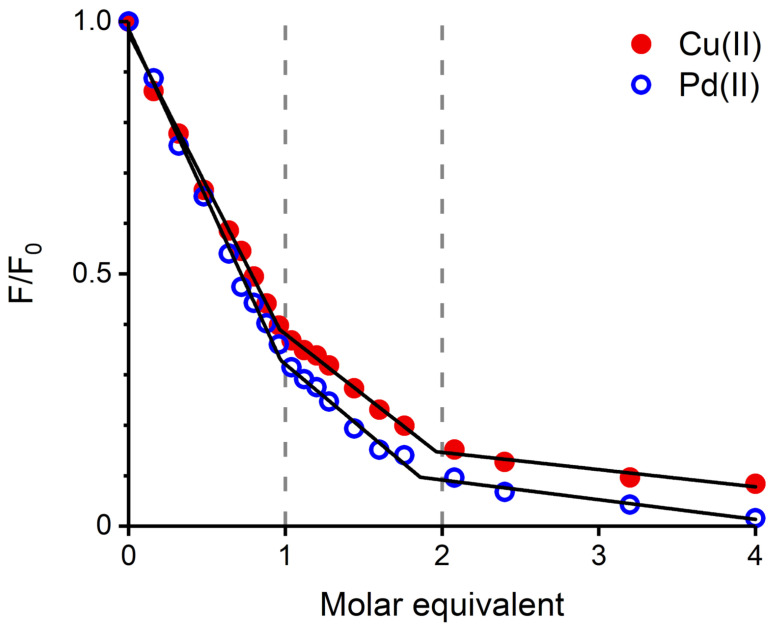
A$\beta_{4-16}$ Tyr10 fluorescence ($\lambda_{ex}$ = 280 nm, $\lambda_{em}$ = 303 nm) quenching by Cu(II) (red dots) and Pd(II) (blue circles). Regions corresponding to the binding of the first and second metal ion equivalents are marked by dashed lines. [Aβ] = 25 μM, [MES] = 20 mM, pH 6.5.

**Figure 4 ijms-21-09200-f004:**
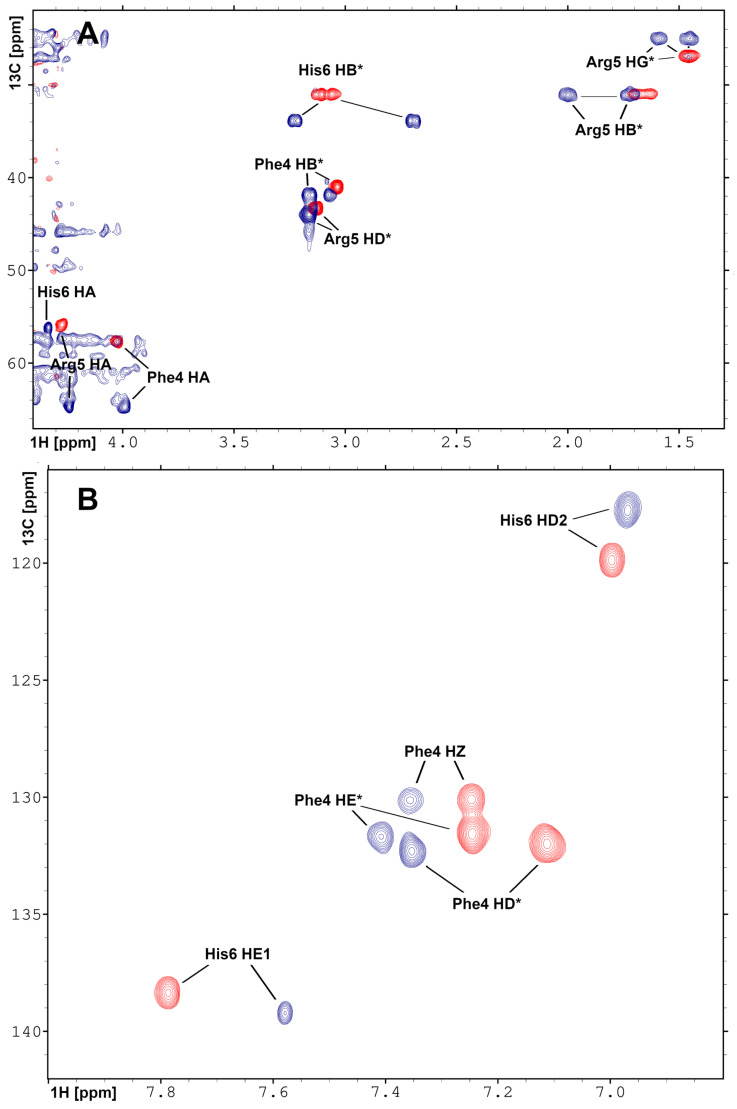
Overlay of the 2D heteronuclear ${}^{1}$H-${}^{13}$C HSQC NMR spectra recorded for (**A**) aliphatic and (**B**) aromatic regions for A$\beta_{4-6}$ peptide in apo (red) and Pd(II) saturated (blue) forms. The assignments and changes in the position of the resonances are shown. The experiments were performed on Varian Inova 500 NMR spectrometer.

**Figure 5 ijms-21-09200-f005:**
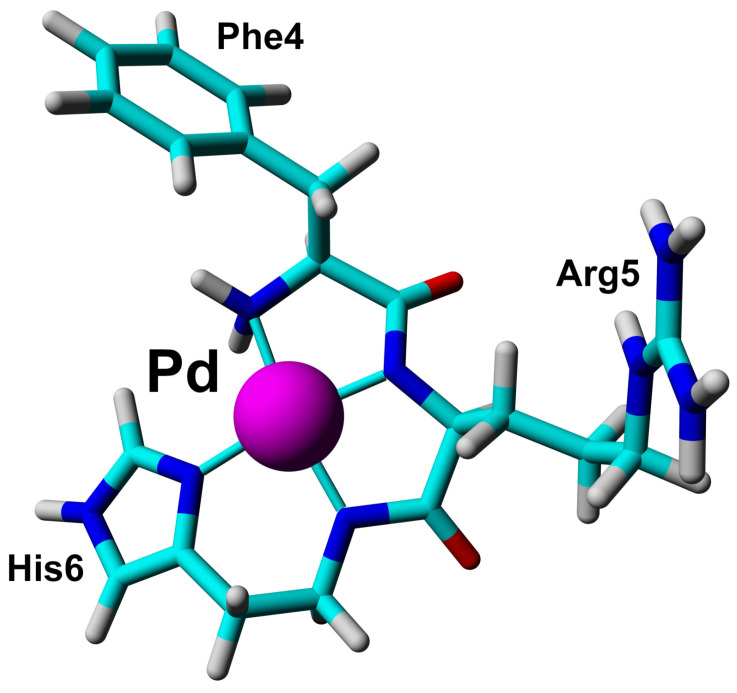
The 3D structure of Phe-Arg-His-amide (FRH) peptide represented the N-terminal ATCUN/NTS motif saturated with Pd(II) ion based on collected NMR constraints and crystallographic data available for GGH tripeptide [33].

**Figure 6 ijms-21-09200-f006:**
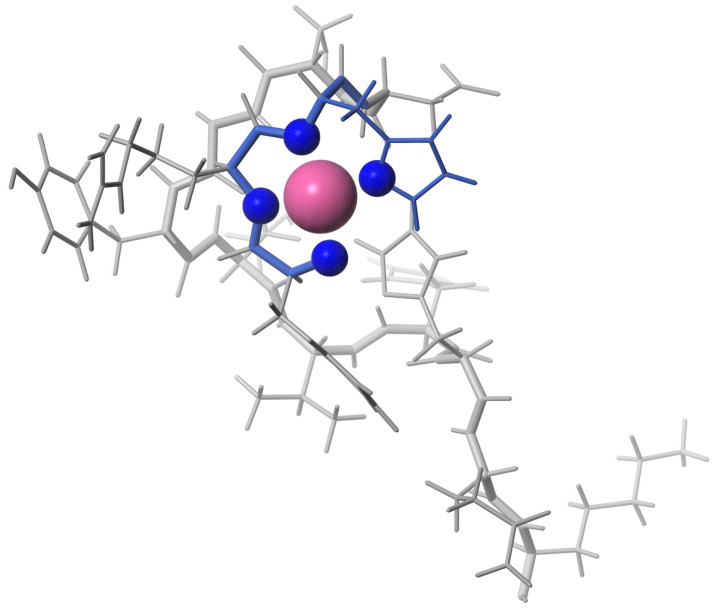
The 3D structure of Pd(A$\beta_{4-16}$) complex included the N-terminal ATCUN/NTS motif binding the Pd(II) ion.

**Figure 7 ijms-21-09200-f007:**
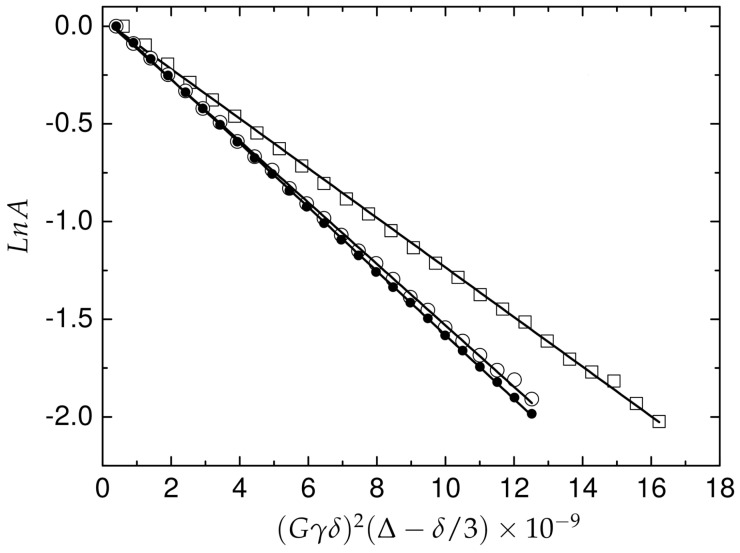
Integral attenuation vs. gradient amplitude in PGSE NMR experiment for A$\beta_{4-16}$ and Pd(A$\beta_{4-16}$) complex in solution. The apo A$\beta_{4-16}$ peptide (filled circles), equimolar Pd(A$\beta_{4-16}$) complex 1:1 (open circles) and 1:1.4 (open squares). The experiments were performed on a Varian Inova 500 NMR spectrometer.

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
