# Peer review of "The Palladium(II) Complex of Aβ4−16 as Suitable Model for Structural Studies of Biorelevant Copper(II) Complexes of N-Truncated Beta-Amyloids"

_ijms, 2020, doi:10.3390/ijms21239200_

Round 1

Reviewer 1 Report

This study by Mital et al. determines the solution state NMR structure of the Pd(II)-Ab4-16 complex. The use of Pd(II) was an interesting and well-justified substitute for Cu(II), which is paramagnetic and would impede structural characterization by NMR. The biological significance of this study lies in the redox-silent nature of the Cu(II)-Ab4-42 complexes compared to redox-active Cu(II)-Ab1-42 species. The latter has been proposed to cause ROS toxicity, while the former might be a scavenger. Overall, the study is sound and the detailed methods regarding the formation of the Pd(II)-peptide complexes could be useful for other Cu(II)-binding proteins.

Minor comments/corrections:

1 . Figure 1: missing labels A, B, and C on the respective panels

2. Figure 2: the text size for the mass-spectra is too small

3. Additional details are needed for the XPLOR calculation. I.e., torsion dynamics, simulated annealing, refinement parameters, restraints etc. TALOS-generated torsion angles should also be tabulated since they appear to be the only restraints.

Author Response

Points 1 and 2. Figures 1 and 2 in the main text were corrected accordingly.

Point 3. We provide additional information about the procedure of structural evaluation of both (apo and Pd(II)-loaded) forms of Aβ4-16 peptide in the ‘Materials and Methods’ subsection ‘Assignment of the 1H, 13C and 15N resonances and 3D structure evaluation of Aβ4-16 peptide and Pd(Aβ4-16) complex in solution’. In addition, the two tables contained restraints generated by TALOS-N software were included in Supporting Materials - Table S5 and S6.

Reviewer 2 Report

To address the broader lines/low resolutions induced by the paramagnetic ions, the authors substituted the Cu(II) with Pd(II) in Abeta4-x peptide complex, in hoping the minimum impact of the substitution could have to the structure features. This is a commonly accepted method if extra caution is taken. Besides characterization using CD, MS, and HPLC analyses of the PD-complex, the Tyr fluorescence titration experiments demonstrated the isostructural features between Cu and Pd complexes.  Full NMR spectroscopic analyses were carried out on several samples (tripeptide and tripeptide-PD complex, Abeta4-x peptide, and Pd-Abeta4-x peptide complex. The experimental design and data analyses sound.  The results, discussion, and conclusion are reasonable. I recommend to publish this work because it presents a useful and interesting case study for the paramagnetic ion substitution in peptide structural studies. However I would like the authors to address a few questions and comments that I have about this manuscript:

The NH signals are missing for Ph4-Arg5-His6 structural motif in all cases investigated. In fact the NH information of these segment is so important to have a better understanding of the structures (apo or complex). While it is convenient for the authors to claim that the fast exchange was the main reason to these missing these signal, the underline causes may not be so straightforward to be recognized.  Why other NH’s of the same peptide do not experience fast exchange?  Authors should have more in-depth explanation.  For the same reason, I would suggest author to push the water content from 90% to 95% to slow down the exchange.

Relaxation properties (T1 and T2) have been measured for all possible 13C sites. What is the main reason 15N relaxation rates were not measured? Even though the NH protons were not detected for Ph4-Arg-5His6, the 15N relaxation rates of other sites, such as Tyr10, may provide more direct dynamic information about this system.

To claim supramolecular aggregates formed between apo and complexed peptides based only on the data of the complexes of two molar ratios is a bit of arbitrary because the “apparent hydrodynamic radius” calculated from the diffusion coefficients are just very rough estimates. Even the apparent hydrodynamic radius means anything here, there is no way to know if the aggregates are between apo and apo, apo and complex, or complex and complex just relying on the diffusion measurements. More carefully designed diffusion experiments, including more mixtures with a range of molar ratios, may help revealing the features of the supramolecular aggregates

Author Response

  • The lack of HN resonances on 1H-15N HSQC spectrum for Pd(II)FRH and Pd(Aβ4-16) peptides can be explained by coordination of the Pd(II) ion which binds to the ATCUN structural motif. Regarding the HN protons in apo tri-peptide (FRH) we suppose the effect of increased linewidth observed due to fast exchange with water. In apo Aβ4-16 the situation seems more puzzling. Indeed, it is interesting that HN signals from Arg5 and His6 are not observed but other amide protons are. That observation can be an effect of exchange between several states with an exchange rate comparable with spectrometer frequency. We once more checked our experimental data, and we made sure that the absence of these HN depended neither on magnetic field (11.7 or 18.8 T) nor on temperature (we made measurements in the range from 277 up to 303 K). So we have to exclude that possibility. We can propose an explanation that apo peptide has some structural propensity at the fragment 6HSGY10 as demonstrated in Figure S6 (Supporting Materials). In that case, The N-terminal HN resonance lines can broaden due to fast exchange phenomena with solvent. In the future, we will try to perform additional attempts to obtain information about HN chemical shifts for residues in the ATCUN motif utilize (among others) the idea proposed by the reviewer.
  • The 13C relaxation measurements were performed for the FRH tri-peptide (representing the N-terminal ATCUN motif) not for Aβ4-16. For FRH peptide we did not detect the HN signals on 1H-15N HSQC spectrum. In the case of the Aβ4-16 peptide, we have to note that for both apo or Pd-loaded forms the concentration in solution was substantially lower which resulted in a decreased quality of relaxation data (13C and 15N). Bearing in mind that we acquired the data at 11.7 T (Varian Inova 500) without cryogenic probead and on the natural abundance of the 15N and 13C isotopes. In the case of the complex Pd(Aβ4-16) the situation was even worse, due to the existence in solution two (or even three) stable species, which additionally decreased the quality of the experimental data. Nevertheless, we intend to get access to a higher high field NMR spectrometer and perform, at least the 15N relaxation measurements.

  • We agree with the comment in general. Although we do not know how to fully respond to it. We were mainly interested in whether the aggregation process is indeed taking place and it is confirmed by diffusion NMR pretty much like in many other papers by analyzing the changes of the apparent hydrodynamic volumes given that the possible exchange between the aggregates is slow or absent. Regarding the potential exchange between apo-apo or apo-complex forms, it seems that with the existence of dimeric forms, in the case of fast exchange the overall diffusion coefficient would be much faster, i.e. we would observe an averaged value depending on the rate of exchange, which by definition should be faster than the self-diffusion of dimeric form. We believe that dimeric complexes are stable and this type of exchange is slow in the NMR timescale. They rather have a apo+complex character, because there was no manifestation of them in the other samples, although the exact stoichiometry remains to be elucidated. Our intention was to signal that Pd(II) substitution empowers one to study this kind of phenomenon. We added an additional sentence to the text ‘The apparent hydrodynamic volume was calculated according to Stokes-Einstein equation using spherical approximation. Although the morphology of the diffusing species is rather not spherical, the proposed approximation is widely used to follow the aggregation phenomena’ which explained the calculation of hydrodynamic volume according to an analysis of the oligomerization phenomena by PGSE-NMR We included two corresponding references to the text (now references 43 and 44):

    - Macchioni A, Ciancaleoni G, Zuccaccia C, Zuccaccia D. Determining accurate molecular sizes in solution through NMR diffusion spectroscopy. Chemical Society Reviews. 2008;37(3):479-89.

    - Taube M, Pietralik Z, Szymanska A, Szutkowski K, Clemens D, Grubb A, Kozak M. The domain swapping of human cystatin C induced by synchrotron radiation. Scientific Reports. 2019;9:8548